# MITTY: DIFFUSION-BASED HUMAN-TO-ROBOT VIDEO GENERATION

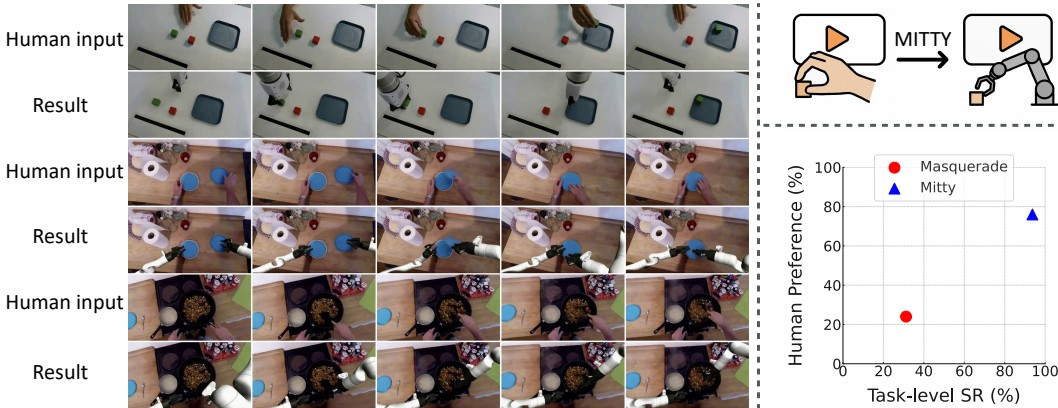

Figure 1: We propose Mitty, a paradoxical in-context learning–based video generation method built on Diffusion Transformers. It can convert human demonstration videos into robotic manipulation videos and achieves high task success rates.

## ABSTRACT

Robots that can learn directly from human demonstration videos promise scalable cross-task and cross-environment generalization, yet existing approaches rely on intermediate representations such as keypoints or trajectories, losing critical spatio-temporal detail and suffering from cumulative error. We introduce Mitty, a Diffusion Transformer framework that enables video In-Context Learning for end-to-end human-to-robot video generation. Mitty leverages the powerful visual and temporal priors of the pretrained Wan 2.2 video model, compressing human demonstration videos into condition tokens and fusing them with robot denoise tokens through bidirectional attention during diffusion. This design bypasses explicit action labels and intermediate representations, directly translating human actions into robotic executions. We further mitigate data scarcity by synthesizing high-quality paired videos from large egocentric datasets. Experiments on the Human-to-Robot and EPIC-Kitchens datasets show that Mitty achieves state-of-the-art performance, strong generalization to unseen tasks and environments, and new insights for scalable robot learning from human demonstrations.

## 1 INTRODUCTION

Humans excel at rapidly acquiring new skills by observing others. If robots could directly learn manipulation policies from a single human demonstration video and generate corresponding robot-execution videos, this would provide a critical path toward cross-task and cross-environment generalization. Yet achieving this has long been a highly challenging goal in robotics.

Existing approaches typically rely on intermediate representations such as keypoints, trajectories, or depth maps to bridge human and robot videos. They first extract keypoints or trajectories from the human demonstration and then condition a rendering module to synthesize robot execution videos.

While intuitive, this approach fails to fully exploit the rich information embedded in demonstration videos and struggles to capture the fine-grained spatio-temporal dynamics essential for robust generalization. Moreover, errors accumulated in the intermediate estimation stage can further degrade performance. This raises a natural question: can we bypass intermediate representations and directly achieve end-to-end human-to-robot video generation?

This task presents several key challenges: (1) Appearance and scene consistency—the generated robot video must match the scene of the human demonstration while preserving a stable, plausible robot embodiment; (2) Action and strategy alignment—the robot's actions must follow the human demonstration yet adapt to structural differences between human hands and robot arms; (3) Data scarcity—despite abundant human and robot videos separately, finely aligned human–robot video pairs are extremely rare. The only public H2R dataset currently contains just 2,600 video pairs across nine tasks, making it difficult to learn generalizable skills from limited data.

To alleviate this scarcity, we propose an automatic paired-data synthesis pipeline using egocentric human videos. Starting from large-scale human activity datasets such as EPIC-Kitchens, we estimate 3D hand keypoints, remove human hands, and inpaint clean backgrounds. We then map keypoint sequences to robot end-effector poses and render robot arms into the video, producing high-quality human–robot paired videos. This approach bypasses traditional intermediate representations and significantly improves both scale and fine-grained temporal consistency, providing stronger training and generalization capacity for our model.

Building on this foundation, we introduce Mitty, a Diffusion Transformer framework for video In-Context Learning. In-context learning (ICL) has shown promise for fewshot learning, offering data-efficient and rapid adaptation at test time. By simply conditioning on one human demonstrations, ICL can predict robot actions to achieve novel tasks at test time without expensive retraining. Our method conditions directly on human demonstration videos to generate corresponding robot-execution videos in an end-to-end manner, requiring no explicit action labels. Mitty leverages Wan 2.2, a powerful video generation model pretrained on massive natural video corpora, to inherit strong visual and temporal priors. Concretely, we compress the human demonstration into condition tokens via a VAE (kept noise-free) and concatenate them with the robot denoise tokens through a bidirectional attention mechanism during diffusion, enabling cross-domain action translation. Mitty supports two inference modes—first-frame-controlled and zero-frame generation—offering greater flexibility in deployment. We further conduct a systematic evaluation across models and settings to provide actionable insights for the community.

Across both the Human-to-Robot dataset and EPIC-Kitchens, Mitty significantly outperforms existing baselines and demonstrates strong generalization. We summarize our contributions as follows:

1. We propose Mitty, the first end-to-end human-to-robot video generation framework built upon a Video Diffusion Transformer.

2. Technically, we leverage in-context learning to achieve both appearance and scene consistency as well as action consistency, significantly improving cross-task generalization.

3. We design an efficient data synthesis strategy and combine it with existing datasets for mixed training, which markedly enhances the model's generalization ability on unseen tasks and environments. Extensive experiments demonstrate the effectiveness and superiority of our approach in terms of generation quality and cross-task consistency.

## 2 RELATED WORKS

### 2.1 VIDEO GENERATION MODELS

Video generation models have evolved rapidly from early GAN-based approaches Pan et al. (2017), UNet-based approaches Guo et al. (2023); Xu et al. (2024); Song et al. (2024) to today's Diffusion Transformer architectures Peebles & Xie (2023); Wan et al. (2025); Zheng et al. (2024); Jiang et al. (2025). Modern Diffusion Transformers can generate high-quality, temporally coherent videos conditioned on text, images, or multi-modal inputs, enabling applications such as controllable video generation Lin et al. (2025); Jiang et al. (2025) and world modeling Gao et al. (2025). Many recent studies also leverage large pretrained video generation models for tasks in robotics and mechani-

cal manipulation Fu et al. (2025), highlighting their potential for cross-domain generalization and interactive learning.

## 2.2 LEARNING FROM HUMAN VIDEOS

A growing body of work investigates how large human-centric video datasets can be used to improve robot policy learning Xie et al. (2025); Shah et al. (2025). Compared to costly and time-consuming teleoperation, large-scale human videos provide a scalable and diverse source of demonstrations. Earlier studies focused on extracting visual representations Chen et al. (2025), deriving reward functions Guzey et al. (2025), or directly estimating motion priors from human videos Wang et al. (2023); Qiu et al. (2025). However, many approaches still rely on additional robot data or specialized hardware such as VR and hand-tracking devices, limiting scalability. Recent progress in 3D hand pose estimation Cheng et al. (2024) helps extract action information directly from RGB videos, but cross-embodiment transfer remains difficult. Humanoid robots can partially alleviate this gap due to their kinematic similarity to humans. Building on these trends, we propose Mitty, which achieves end-to-end generation of robot videos directly from human demonstrations without extracting intermediate representations such as pose, trajectories, or depth, and better leverages the fine-grained details contained in the original human demonstration videos.

## 2.3 IN-CONTEXT LEARNING

In-context learning (ICL) Brown et al. (2020); Alayrac et al. (2022) has demonstrated remarkable capability for adapting models to new tasks at inference time. In the visual generation domain, recent approaches have leveraged ICL to achieve high-quality image generation Huang et al. (2024); Zhang et al. (2025a;b); Song et al. (2025); Huang et al. (2025); Gong et al. (2025) and video generation Zhang et al. (2024); Kim et al. (2025); Yu et al. (2025). In robotics, preliminary studies Shah et al. (2025) have explored applying ICL to visuomotor policies using either teleoperation or simulation data. However, these methods are constrained by data collection costs and limited task diversity, making large and heterogeneous datasets essential for effective adaptation. We adopt an In-Context Learning framework built on the Wan 2.2 video diffusion model to translate human demonstration videos into robot-arm executions, ensuring visual and action consistency throughout generation.

## 3 METHOD

In this section, we first define our problem formulation and pverall architecture in Sec. 3.1, then we describe in detail how video in-context learning is achieved via bidirectional attention in Sec. 3.2, and finally explain our synthetic paired-data construction pipeline in Sec. 3.3.

### 3.1 OVERALL ARCHITECTURE

We formulate human-to-robot video generation as a conditional denoising problem. Given paired data consisting of a human demonstration video $V^H = \{v_1^H, \ldots, v_N^H\}$ and the corresponding robot execution video $V^R = \{v_1^R, \ldots, v_N^R\}$, our objective is to model the conditional distribution $p_\theta(V^R|V^H)$ that captures fine-grained spatio-temporal correspondences between human actions and robot executions. We consider two settings: (i) **H2R**(Human-to-Robot Video Generation), where the model directly generates a robot execution video from a human demonstration without providing any initial robot frame; and (ii) **HI2R**(Human-and-Initial-Image-to-Robot Video Generation), which extends H2R by additionally supplying an initial robot frame to define the robot's initial state and guide embodiment and motion planning.

We implement this formulation in a single unified framework built upon Wan 2.2 Wan et al. (2025), a state-of-the-art diffusion-based video generation model pretrained on large-scale natural videos. Both human and robot videos are encoded into latent tokens using the same VAE-based video encoder. Human latents act as clean conditioning tokens, while robot latents act as denoising targets. These tokens are concatenated along the temporal dimension and fed into a Diffusion Transformer enhanced with bidirectional attention, enabling information to flow between modalities at each denoising step. This unified design supports both zero-frame generation (H2R) and first-frame-

conditioned generation (HI2R), sharing parameters and priors across both settings while allowing fine-grained control over the robot's initial state and stable motion planning across diverse tasks.

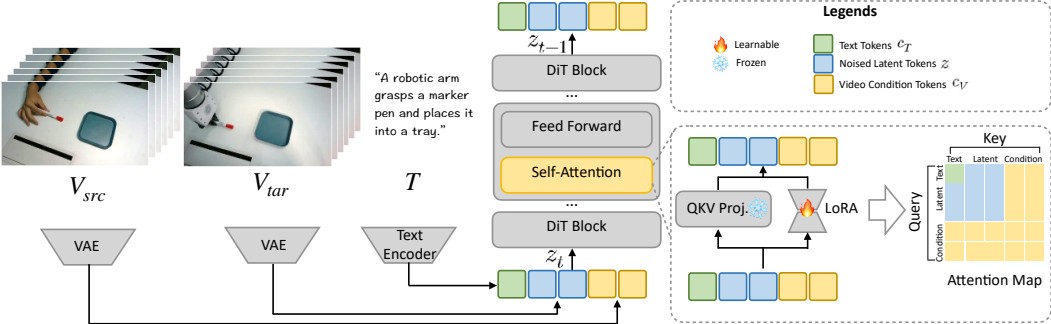

Figure 2: Overall architecture of Mitty. We build Mitty on a Diffusion Transformer–based video generation model and employ an In-Context Learning paradigm. The human demonstration video (input) and the noisy robot video latents (denoise stream) are concatenated, with noise injected only into the robot branch. A bidirectional attention mechanism enables cross-modal information flow, allowing the model to learn to generate robotic videos directly from human operation demonstrations.

## 3.2 VIDEO IN-CONTEXT LEARNING VIA BIDIRECTIONAL ATTENTION

To achieve cross-domain video in-context learning between human and robot modalities, we enhance the Diffusion Transformer with a **bidirectional attention mechanism** linking human-condition tokens and robot-denoise tokens. This allows the model to dynamically align temporal cues, motion patterns, and object interactions across domains while leveraging the strong visual–temporal priors from the pretrained video backbone.

**Diffusion Process and Noise Injection.** Let $\mathbf{z}_0^R = \text{VAE}_{\text{enc}}(\mathbf{V}^R)$ denote the robot video latent. During training, we progressively add noise only to the robot latents while keeping the human latents clean:

$$\mathbf{x}_t^R = \sqrt{\bar{\alpha}_t}\,\mathbf{z}_0^R + \sqrt{1 - \bar{\alpha}_t}\,\boldsymbol{\epsilon}, \quad \boldsymbol{\epsilon} \sim \mathcal{N}(\mathbf{0}, \mathbf{I}). \tag{1}$$

with cumulative noise schedule

$$\bar{\alpha}_t = \prod_{s=1}^{t} \alpha_s, \quad t \in \{1, \dots, T\}. \tag{2}$$

This setup enables us to model the conditional distribution $p_\theta(\mathbf{V}^R \mid \mathbf{V}^H)$ without requiring explicit action or trajectory labels.

**Token Representation and Embeddings.** Let $\mathbf{z}_0^H = \text{VAE}_{\text{enc}}(\mathbf{V}^H)$ denote the human video latent. Tokens are formed as

$$\mathbf{C} = \mathbf{z}_0^H + \mathbf{E}_{\text{time}} + \mathbf{E}_{\text{mod}(h)}, \qquad \mathbf{D} = \mathbf{x}_t^R + \mathbf{E}_{\text{time}} + \mathbf{E}_{\text{mod}(r)}. \tag{3}$$

Here $d$ denotes the token/channel dimension and $\mathbf{E}_{\text{time}}, \mathbf{E}_{\text{mod}(\cdot)}$ are temporal and modality embeddings.

**Bidirectional Attention Coupling.** At each layer, we exchange information in both directions (row-wise softmax):

$$\tilde{\mathbf{C}} = \text{Softmax}\left(\frac{\mathbf{C}\mathbf{D}^\top}{\sqrt{d}}\right)\mathbf{D}, \qquad \tilde{\mathbf{D}} = \text{Softmax}\left(\frac{\mathbf{D}\mathbf{C}^\top}{\sqrt{d}}\right)\mathbf{C}. \tag{4}$$

The updated tokens $[\tilde{\mathbf{C}}; \tilde{\mathbf{D}}]$ are concatenated along the token dimension and fed to subsequent Transformer blocks.

**Denoising and Reverse Update.** The network predicts $\boldsymbol{\epsilon}_\theta(\mathbf{x}_t^R, \mathbf{C}, t)$ on the robot branch and performs

$$\mathbf{x}_{t-1}^R = \frac{1}{\sqrt{\alpha_t}}\left(\mathbf{x}_t^R - \frac{1 - \alpha_t}{\sqrt{1 - \bar{\alpha}_t}}\,\boldsymbol{\epsilon}_\theta(\mathbf{x}_t^R, \mathbf{C}, t)\right) + \sigma_t \mathbf{z}, \quad \mathbf{z} \sim \mathcal{N}(\mathbf{0}, \mathbf{I}), \tag{5}$$

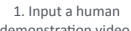 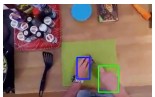 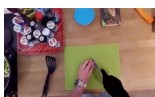 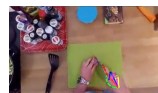 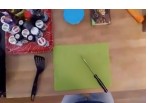 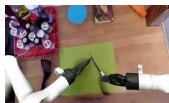

| 1. Input a human demonstration video. | 2. Detect hands using Detectron 2. | 3. Segment hands and arms using SAM 2. | 4. Detect hand keypoints. | 5. Inpaint the removed hand regions. | 6.Rendering the robot arms. |

Figure 3: Starting from a human demonstration video, we first detect hands using Detectron2 and then segment hands and arms using Segment Anything. Next, we perform hand keypoint detection and inpaint the removed hand regions to obtain clean background frames. We then apply inverse kinematics solving to map the detected hand keypoints to robot arm poses and render the robot arms into the videos. Finally, with a human-in-the-loop filtering process, we curate over 6,000 high-quality synthetic human–robot paired videos to support the training of our Mitty model.

with $\sigma_t$ given by the variance schedule. The final video is

$$\hat{\mathbf{V}}^R = \text{VAE}_{\text{dec}}(\mathbf{z}_0^R). \tag{6}$$

This models the conditional generation without action labels, and supports both *H2R (Zero-frame)* and *HI2R (First-frame conditioned)* modes defined in Sec. 3.1, enabling either generation from human demonstrations or controlled execution with an initial robot frame.

### 3.3 DATASET CONSTRUCTION

A key bottleneck in robotic learning lies in data acquisition: collecting real-world robot manipulation data is costly and slow, which limits generalization across large-scale tasks. Meanwhile, ego-centric human activity datasets such as EPIC-KitchensDamen et al. (2020), Ego4DGrauman et al. (2022), and EgoExo4DGrauman et al. (2024) have accumulated millions of high-quality demonstrations covering diverse actions and environments. Effectively transferring these large-scale human videos into robotic learning is critical to overcoming the current data bottleneck.

To alleviate the scarcity of human–robot paired videos, we build upon the data rendering approach proposed in the MasqueradeLepert et al. (2025) paper and introduce an automated pipeline. This pipeline takes egocentric human videos as input and produces robot-arm rendered results through the following steps.

**Hand Pose Estimation:** We use models such as HaMeR for 3D Hand Mesh Recovery to extract 3D hand keypoints and motion trajectories from ego-centric videos.

**Hand Segmentation and Removal:** We first use Detectron2 Wu et al. (2019) to detect human hands, and then apply Segment Anything 2 (SAM2) Ravi et al. (2024) to perform fine-grained segmentation and remove the detected hands and forearms from the video.

**Video Inpainting:** We apply E2FGVILi et al. (2022), a video inpainting model to fill the removed regions across frames, producing clean background videos without hands.

**Pose Mapping:** The predicted hand keypoints are mapped to robot end-effector poses, including target position (midpoint between thumb and index finger), target orientation (plane normal plus fitted vector), and gripper opening (thresholded thumb–index distance).

**Robot Arm Rendering:** Using RobotSuite Zhu et al. (2020), we render robot arms corresponding to the mapped poses into the inpainted videos. Fine-tuning of poses and data cleaning/filtering further improves the fidelity of the resulting paired videos.

Given the multi-step nature of our automated data generation pipeline, cumulative errors and inconsistencies can arise across segments. To mitigate these issues, we employ a human-in-the-loop filtering mechanism to rigorously audit and remove low-quality samples, thereby enhancing data fidelity and internal consistency. After filtering, each resulting video is segmented into clips of 81 frames sampled at equal intervals, yielding approximately 6,000 video clips that form our training and testing sets. This procedure produces a high-quality human–robot paired dataset that furnishes robust training support for In-Context Diffusion Transformer models such as Mitty and provides a solid foundation for reliable cross-task and cross-environment generalization.

## 4 EXPERIMENTS.

### 4.1 SETUP.

We build on the pretrained Wan 2.2 TI2V-5B dense model and additionally train on the Wan 2.2 TI2V-14B MoE model. We adopt a LoRA-based fine-tuning strategy, simultaneously adapting both high-noise and low-noise branches. The TI2V-5B model is trained for 20k steps, while the larger TI2V-14B model is trained for 5k steps due to computational cost considerations. The LoRA rank is set to 96 with a fixed learning rate of $1 \times 10^{-4}$. All experiments are conducted on two H200 GPUs. Both training and inference operate on 81 frames per clip at a resolution of 416×224, with an effective batch size of 4.

### 4.2 DATASETS AD BENCHMARK.

We evaluate Mitty on two primary datasets. Human2Robot (H2R) Xie et al. (2025)contains 2,600 paired human–robot videos collected via VR teleoperation, covering diverse manipulation tasks with fine-grained temporal alignment. We exclude videos shorter than 81 frames, resulting in 1,019 videos for training and 255 videos for testing. EPIC-KITCHENS Damen et al. (2020) is a large-scale egocentric kitchen dataset. Using our pipeline in Sec. 3.3, we render robot arms into kitchen scenes, producing 5,373 training videos and 597 testing videos. All quantitative evaluations reported in this paper are computed on the respective held-out test sets to ensure fair and consistent benchmarking across datasets.

### 4.3 METRIC.

We evaluate performance with two criteria. First, video quality is assessed using standard metrics—Fréchet Video Distance (FVD), PSNR, MSE, and SSIM—and further evaluated on the test sets of both datasets. Second, task success rate serves as our primary metric: we predefine failures as cases with obvious visual artifacts, temporal discontinuities or distortions, or incorrect robot-arm motions. Three human experts independently review and score each generated video; disagreements are resolved via discussion to reach consensus.

### 4.4 BASELINE METHODS.

All baselines are instantiated on the Wan 2.2 TI2V framework. Our primary setup uses the TI2V-5B backbone with first-frame conditioning and human reference videos, which forms the core configuration of Mitty. To study model scaling, we also evaluate a larger T2V-14B configuration under the same setting.

We then consider the ablated variants used in the application study: (i) *first-frame only* — the model predicts subsequent frames using only the initial robot frame and the task description (the human reference video is removed); (ii) *text-free* — the task description is removed while keeping the initial robot frame and the human reference video. Finally, we assess training strategies by comparing *separate training* (a dedicated model per dataset) versus *mixed training* (joint training on both datasets).

Table 1: Video generation quality metrics across Human2Robot and EPIC-Kitchens datasets. Lower FVD and MSE indicate better quality, while higher PSNR, SSIM, and SR (Success Rate, %) indicate better reconstruction fidelity and task performance. The best results are highlighted in bold.

| Dataset | Method / Setting | FVD↓ | PSNR↑ | SSIM↑ | MSE↓ | SR↑ |
|---------|------------------|------|-------|-------|------|-----|
| Human2Robot | TI2V 5B (w/o first frame) | 102.4 | 21.5 | 0.835 | 0.00837 | 87.8 |
| | TI2V 5B (w first frame) | 90.2 | 21.7 | 0.837 | 0.00806 | 90.6 |
| | T2V 14B | **87.0** | **22.7** | **0.851** | **0.00649** | **93.7** |
| EPIC-Kitchens | TI2V 5B (w/o first frame) | 396.6 | 14.9 | 0.686 | 0.0399 | 79.7 |
| | TI2V 5B (w first frame) | 301.2 | 14.5 | 0.682 | 0.0405 | 83.4 |
| | T2V 14B | **290.5** | **15.7** | **0.693** | **0.0326** | **86.3** |

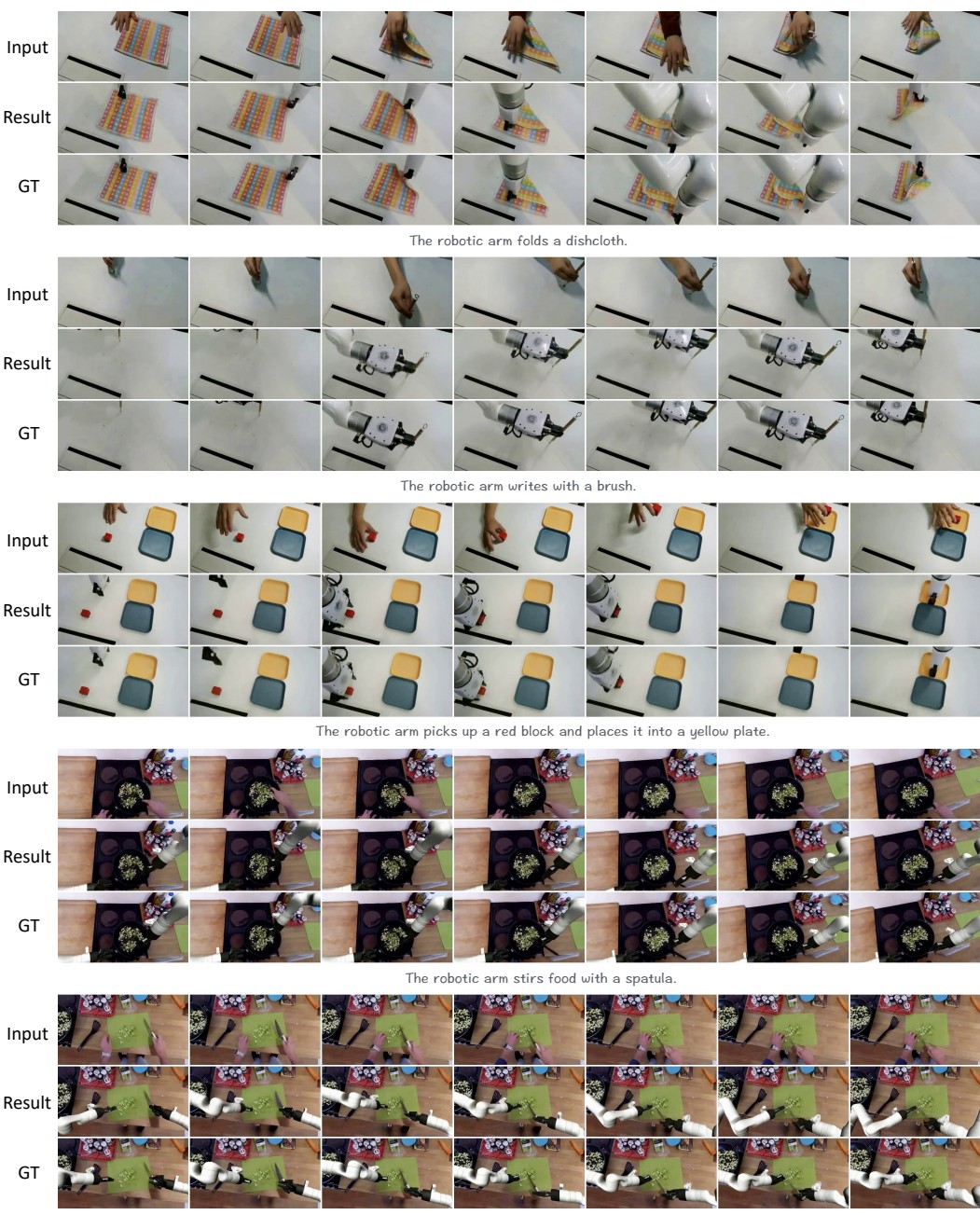

Figure 4: Mitty's generation results on Human2Robot and EPIC-Kitchens datasets. In each group of results, the first row shows the human demonstration videos, the second row shows the outputs generated by our method, and the third row shows the ground-truth robot execution videos.

## 4.5 RESULTS.

**Quantitative Evaluation**

Figure 5 shows Mitty's qualitative results on Human2Robot and EPIC-Kitchens. Each group contains three rows: the first row is the human demonstration, the second row is Mitty's zero-frame generation without first-frame conditioning, and the third row is the ground truth robot-execution video. We observe that Mitty accurately preserves scene layout and object interactions while producing smooth, temporally coherent robot motions. Thanks to its In-Context Learning design, Mitty

Figure 5: Comparison with Masquerade. Masquerade uses a multi-stage pipeline—detecting human hand joint chains, implanting hands, and rendering robot arms—which is prone to cumulative errors such as implanting failures or inaccurate joint detection, resulting in lower success and usability rates. In contrast, our approach is a streamlined end-to-end method that directly maps human demonstrations to robot executions, achieving higher reliability and consistency. The red boxes highlight the regions where Masquerade encounters problems.

also generalizes robustly to unseen tasks and environments, maintaining strong visual consistency, action consistency, and background stability.

**Qualitative Results**

Table 1 summarizes our results on the Human2Robot and EPIC-Kitchens datasets. Across both datasets, adding the first-frame condition consistently reduces FVD and MSE while slightly increasing PSNR, SSIM, and SR, demonstrating more stable and faithful video generation. On Human2Robot, our larger T2V 14B model achieves the best overall performance, yielding the lowest FVD and MSE and the highest PSNR, SSIM, and SR compared to TI2V 5B. In contrast, the EPIC-Kitchens dataset presents more diverse scenes, more complex environments, and moving camera viewpoints, which make the task significantly more challenging. Consequently, performance metrics on EPIC-Kitchens are generally lower than on Human2Robot, reflecting the increased difficulty of achieving high-fidelity generation under such conditions.

Table 2: Ablation study on Human2Robot and EPIC-Kitchens datasets under three settings: (1) w/o ref video, (2) w/o task description, and (3) Full model with separate or mixed training. Lower FVD and MSE indicate better perceptual and reconstruction quality, while higher PSNR, SSIM, and SR indicate better fidelity. The best results are highlighted in bold.

| Dataset | Method / Setting | FVD↓ | PSNR↑ | SSIM↑ | MSE↓ | SR↑ |
|---|---|---|---|---|---|---|
| Human2Robot | w/o ref video | 237.6 | 22.0 | 0.858 | 0.0091 | 64.7 |
| | w/o task description | 94.2 | 21.4 | 0.837 | 0.0091 | 87.5 |
| | Full (Mixed training) | 255.4 | 16.6 | 0.742 | 0.0238 | 72.2 |
| | Full (Separate training) | **90.2** | **21.7** | **0.837** | **0.0081** | **90.6** |
| EPIC-Kitchens | w/o ref video | 381.4 | 13.6 | 0.463 | 0.0528 | 63.5 |
| | w/o task description | 364.1 | 14.5 | 0.682 | 0.0398 | 81.1 |
| | Full (Mixed training) | 377.4 | 13.9 | 0.664 | 0.0467 | 71.2 |
| | Full (Separate training) | **301.2** | **14.6** | **0.689** | **0.0405** | **83.4** |

## 4.6 ABLATION STUDY.

Table 2 presents the ablation results on the Human2Robot and EPIC-Kitchens datasets using the TI2V-5B model. Considering the additional training and inference cost, we adopt the TI2V-5B with

Table 3: Comparison between our method and Masquerade on task-level SR (Success Rate) and human preference. The best results are highlighted in bold.

| Method | Task-level SR (%) | Human Preference (%) |
|---|---|---|
| Masquerade | 31.2 | 23.7 |
| Ours | **93.7** | **76.3** |

first-frame conditioning as our default baseline. When the human reference video is removed, the model predicts subsequent frames using only the initial robot frame and task description, resulting in clear degradation in FVD, PSNR, SSIM, and SR on both datasets. In contrast, removing task description prompts causes only minor changes, indicating that Mitty relies more on visual demonstrations than textual cues. Notably, the impact of removing the human reference video is more severe on EPIC-Kitchens due to its more diverse scenes and moving camera viewpoints, further emphasizing the importance of strong visual conditioning under complex environments. Finally, because the two datasets differ substantially in tasks and environments (e.g., single-arm vs. dual-arm manipulation and varying scene complexity), the full model trained separately on each dataset outperforms mixed training.

### 4.7 COMPARE WITH MASQUERADE.

Masquerade employs a multi-stage pipeline—hand segmentation and pose estimation, background inpainting, and robot-arm rendering—that leverages large-scale human videos but accumulates errors at each step. Typical failure cases include inaccurate hand or pose predictions, limited robot-arm reach due to rendering constraints, missing depth cues that confuse hand–object relationships, and incomplete inpainting of accessories such as watches that leaves artifacts. These issues lead to unrealistic or misaligned robot motions. In contrast, our approach is end-to-end, directly mapping human demonstrations to robot executions without intermediate steps. This streamlined design produces more coherent and plausible robot behaviors. For human preference evaluation, three independent experts each reviewed and scored 100 generated video samples, expressing their preference between our method and Masquerade. The final scores were obtained by aggregating their independent judgments. Table 3 shows that our method substantially outperforms Masquerade in both task-level SR and human preference.

## 5 LIMITATION AND FUTURE WORK.

Although Mitty demonstrates strong performance and generalization, it still has several limitations. At present, Mitty can only generate robot-arm execution videos and cannot explicitly predict action sequences, which limits its direct applicability to real-world robot policy execution. Future work will explore integrating action prediction into the framework, leveraging larger-scale real human demonstration data, and extending to more complex tasks and embodiments to improve long-term temporal consistency and real-world applicability.

## 6 CONCLUSION

We presented Mitty, a Diffusion Transformer framework enabling in-context learning for end-to-end human-to-robot video generation. Leveraging Wan 2.2 and a paired-data synthesis pipeline, Mitty bypasses intermediate representations and directly translates human demonstrations into robot videos. Experiments on Human-to-Robot and EPIC-Kitchens show state-of-the-art performance and strong generalization. We also conducted comprehensive ablation studies across multiple variants, providing quantitative evidence and actionable insights into the contribution of each component. Furthermore, we compared Mitty with rendering-based approaches such as Masquerade: unlike their multi-stage pipelines prone to accumulated errors, Mitty's streamlined end-to-end design produces smoother, temporally coherent, and more reliable robot motions. These results demonstrate the potential of diffusion-based in-context learning for scalable, robust robot learning from human videos.

## CODE OF ETHICS

The authors have read and acknowledge adherence to the ICLR Code of Ethics.

## ETHICS STATEMENT

All datasets used in this work are publicly available and widely adopted in the research community. We comply with dataset licenses and usage guidelines. Human figures appear only as part of these existing benchmarks to evaluate generalization across diverse visual domains. No private or newly collected human data was used.

## REPRODUCIBILITY STATEMENT.

All datasets, model configurations, and training details used in this work are described in the paper. We will release the synthetic paired human–robot dataset, model checkpoints, and inference scripts upon publication to facilitate full reproducibility. Hyperparameters, architecture details, and evaluation metrics are explicitly documented. We also provide ablation studies to clarify the effect of each component. Together, these measures ensure that researchers can replicate and extend our results without ambiguity.

## USE OF LARGE LANGUAGE MODELS

We only used large language models such as GPT-4 and GPT-5 to assist with English grammar refinement and error correction at the writing stage. All technical content—including method design, experimental setup, and quantitative results—was independently conceived, implemented, and verified by the authors. Large language models were not used to modify any experimental data or code. This guarantees the scientific integrity and originality of this work.

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

# A    APPENDIX

In the appendix, we provide extended qualitative results and additional video demonstrations to complement the figures and tables in the main paper. For anonymity, we host all videos on two anonymous links. The original full-resolution videos are included in the supplementary material to allow reviewers to inspect the generated results in detail.

**Video 1** showcases our Mitty model's generated robot-execution videos across Human2Robot and EPIC-Kitchens, illustrating its visual consistency, action coherence, and robustness across diverse tasks.

**Video 2** presents a side-by-side comparison between Mitty and Masquerade, enabling a direct qualitative assessment of differences in motion quality, temporal stability, and scene fidelity. From top to bottom, the three rows correspond to the human demonstration video, the Masquerade results, and our Mitty results.

The anonymous links are:

- Anonymous Link 1 (Mitty results): `https://limewire.com/d/bCrCQ#D4lsgOngJZ`
- Anonymous Link 2 (Mitty vs. Masquerade): `https://limewire.com/d/2OwUH#qVP5tFne6A`

