# OpenReview forum: "Mitty: Diffusion-based Human-To-Robot Video Generation"
_ICLR.cc/2026/Conference — ICLR 2026 Conference Withdrawn Submission_

### Official Review · Reviewer_tuGg · 2025-10-30

**Soundness:** 4
**Presentation:** 3
**Contribution:** 4
**Rating:** 6
**Confidence:** 3

**Summary:**

This paper introduces an end-to-end method for human-to-robot video generation using in-context learning on a diffusion transformer. The qualitative results show great performances in beta generation metrics, while the human preference results greatly exceed stated our existing work.

**Strengths:**

This work makes good usage of existing pre-trained video model, and proposes an effective conditional mechanism using bidirectional attention for video generation. Dataset construction, experiment design, and baseline design are done well, with the resulting empirical performances being very strong.

This paper is written well and tackles an important robot learning problem: Solving the problem of paired human-robot video generation can have a great impact on the field, and the results shown indicate a good step towards solving this problem.

**Weaknesses:**

While the end-to-end nature of this work seems likely to lead to less compounding errors in video generation compared to a multistage approach like Masquerade. The human preference rating results seem a bit unfair for Masquerade because this work uses a human-in-the-loop filtering step while Masquerade does not.

For the sake of improving this paper, I would be interesting to see the same set of experiments but without the human-in-the-loop filtering step. But for designing a useful pipeline, I fully acknowledge that the filtering step is probably required.

**Questions:**

- line 141 typo on "pverall"

---

### Official Review · Reviewer_ghkP · 2025-10-31

**Soundness:** 3
**Presentation:** 3
**Contribution:** 2
**Rating:** 2
**Confidence:** 4

**Summary:**

This paper introduces Mitty, a Diffusion Transformer framework designed for end-to-end human-to-robot video generation — translating human demonstration videos directly into robot-execution videos. Unlike prior work relying on intermediate representations such as keypoints, trajectories, or depth maps, Mitty enables video-level in-context learning (ICL) that captures fine-grained spatio-temporal relationships between human actions and robot motions.

**Strengths:**

1. Propose a unified diffusion-based transformer that learns to map human demonstration videos to robotic executions without explicit action labels or intermediate representations, enable end-to-end video generation.
2. Propose a synthetic paired dataset creation pipeline which automatically constructs high-quality human–robot video pairs by rendering robot arms into egocentric human videos, using hand segmentation, inpainting, and pose-mapping. This pipeline can leverages egocentric human videos for robot learning.
3. Desmonstrate that foundation video models (Wan 2.2) can serve as priors for robotic understanding and strong generalization.

**Weaknesses:**

1. Diffusion Transformer novelty is somewhat incremental, largely built upon Wan 2.2 with modifications (LoRA tuning and bidirectional attention). While effective, it may not constitute a deep model innovation.
2. The synthetic paired data generation pipeline introduces possible domain biases, accumulated errors at each step may lead to unrealistic or misaligned robot motion videos.
3. Evaluation focuses mainly on visual quality and human preference, lacking explicit quantitative task success metrics in real-world settings.

**Questions:**

1. The synthetic paired data generation pipeline appears to follow similar steps as Masquerade, and the ground-truth paired dataset seems to have comparable quality. Given this, what factors contribute to Mitty’s significantly better generation performance compared to Masquerade?
2. Beyond video generation metrics, are there any experiments demonstrating that the generated robot-execution videos can be effectively used for real-world robot policy learning or execution? Without evidence of practical applicability, it is difficult to assess the true impact of the proposed approach on robot learning.

---

### Official Review · Reviewer_M7LE · 2025-11-01

**Soundness:** 3
**Presentation:** 3
**Contribution:** 2
**Rating:** 4
**Confidence:** 4

**Summary:**

This work leverages robot video generation policies to generate robot videos (not trajectories) based on encoded videos of human demonstrations in the same scene. The idea is that eventually these videos will be useful for robot training, but the basic approach (aside from in-painting to remove the human) is video-to-video translation.  One could apply the same technique to have any animated arm or arbitrary concept interact instead so the robotics angle is motivational but not core.  This is also where the evaluation falls short.

**Strengths:**

Conceptually, if a human demonstration could be easily translated into a robot demonstration the idea is that (in principle) we could have arbitrary training data for robotics.

**Weaknesses:**

In practice, this is a video-to-video translation task, with a focus on specific intermediate modules and domains that make it geared towards robotics specifically.  The tasks take the traditional overhead PnP format, but there is no evidence provided that the resulting videos are useful to robotics.

Minor:
- L240 improper citation spacing

**Questions:**

- Please clarify the pretraining required for the diffusion? For example, how does coverage in the training of the model inform what can be used for conditioning?  How much domain shift or visual complexity can be handled? Silly example: a video of knocking over and occluding objects before building a tower.
- What are common failure conditions? When are the arm kinematics impossible? The depth or interaction physically impossible? What causes such failures?
- What benefits does this approach have over other in-painting robotics work (e.g. Bahl 2022) where robot primitives are directly predicted/refined which guarantees executability of a final robot policy?
- Metric: define thresholds for L297

---

### Official Review · Reviewer_tN2V · 2025-11-01

**Soundness:** 2
**Presentation:** 3
**Contribution:** 2
**Rating:** 4
**Confidence:** 3

**Summary:**

The paper presents Mitty, an end-to-end human-to-robot video generator. The key idea is that given a human demo video, the model generates the corresponding robot-manipulation video. The data involves ~6k human-robot video pairs based on egocentric datasets (e.g., EPIC-Kitchens) and inpainting robot-arm for the paired. The paper proposes finetuning pretrained Wan 2.2 Diffusion Transformer with 81 frame clips in context at 416x224 resolution. The results show that Mitty improves over the previous approach, Masquerade in terms of visual assessment of task success and preference by 3 expert human viewers.

**Strengths:**

Mitty addresses an important aspect of the data collection problem for robotics. While Masquerade was a multi-step pipeline with learned components, Mitty shows that it can be learned end-to-end and can improve the visual quality.

**Weaknesses:**

The paper explores the benefits of distilling the human-robot paired data (where the robot is synthetically generated with a pipeline similar to Masquerade). While this is interesting investigation, there are questions on the dataset quality and what its effects would be on the downstream real robot tasks.

A noted limitation is that Mitty only generates robot videos and not actions. While the paper presents improvements in visual quality with end-to-end training, no real robot experiments are performed as there are no closed-loop policies.
This leaves an important open question on whether visually good robot renderings can translate into accurate robot actions.
Some evidence from papers like UniSim (Yang et al) might indicate that's the possible but a fair comparison to Masquerade would be on real robot tasks.

**Questions:**

Can you show the robot rendered synthetic data training can transfer into improvement for real robot tasks?

Why joint training across different human-robot paired datasets doesn't help? It seems one reason to train end-to-end is to distill the commonalities across different domains and capture the invariances (like the background doesn't change unless interacted with, human arms to robot ones)

Can we have more granular the human experts judgement (consistency, self-collisions between arms, occlusion handling, etc.) to better understand what are the gaps in policies learning human-to-robot renderings?

---

### Note · Authors · 2025-11-12

I have read and agree with the venue's withdrawal policy on behalf of myself and my co-authors.